# Development of a standardized consensus lexicon for terms related to micronutrient programs

Lwin Mar Hlaing[1,2]*, Megan W. Bourassa[1], Kenneth H. Brown[1,3], Reed Atkin[1‡], Saskia J. M. Osendarp[1‡], Aishani Gupta[4], Sonja Y. Hess[1,3]

1 The Micronutrient Forum, Washington, DC, United States of America, 2 Ministry of Health, National Unity Government, Myanmar, 3 Institute for Global Nutrition and Department of Nutrition, University of California Davis, Davis, CA, United States of America, 4 Independent Regulatory Expert, Canada

☯ These authors contributed equally to this work.
‡ RA and SJMO also contributed equally to this work.
* lmhlaing78@gmail.com

**Data Availability Statement:** All relevant data are within the manuscript and its Supporting Information files. In addition, all data (Micronutrient-related terms with definitions in the

## Abstract

Inconsistent use of terminology among diverse stakeholders hinders effective communication in micronutrient programs, especially large-scale food fortification (LSFF) which involves stakeholders from different sectors. To align the terminology use, the Micronutrient Data Innovation Alliance (DInA) of the Micronutrient Forum (MNF) created a lexicon of terms related to LSFF and other micronutrient programs. The purpose of this lexicon is to establish a central repository of consensus definitions of key terms to facilitate communication among diverse stakeholders involved in micronutrient programs including public and private sectors, donor agencies, food industries, academic institutions, etc. This paper describes the methodology of lexicon development. Important terms related to micronutrient programs were compiled from multiple sources, including United Nations agencies, program implementation and technical support agencies, relevant websites, and scientific literature. The selection of terms was guided by key micronutrient interventions (fortification, supplementation, dietary diversification) and the program cycle (assessment, planning, implementation, monitoring and evaluation). Definitions of terms were identified from these references and checked for consistency across different sources. For terms with multiple definitions, a modified Delphi method was applied to harmonize the definitions. The first draft lexicon (n = 113 terms) was reviewed by six experts from the University of California, Davis (UCD) and MNF, and second draft (n = 115 terms) was shared with 24 global micronutrient experts for feedback. Fifty-four terms were found to have multiple definitions. Of which, minor modification was made for 12-terms with nominal difference and remaining 42-terms were shared with over 140 micronutrient-experts disseminated via an online survey through newsletters and emails to solicit experts' opinions on the most appropriate definition or a modified one. Nineteen legal terms and 83 micronutrient terms (n = 102 terms) were subsequently added. Overall, 39 experts from diverse areas of expertise (LSFF, micronutrient program planning and implementation, surveys and research, policy development, food industry regulations, food safety, and public health nutrition) participated in the online survey. The terms with

current lexicon) can be accessible in the Micronutrient Forum website through https://dinalexicon.micronutrientforum.org/.

**Funding:** This work was supported by the Bill & Melinda Gates Foundation (Grant number of INV-036678). Under the grant conditions of the Foundation, a Creative Commons Attribution 4.0 Generic License has already been assigned to the Author Accepted Manuscript version that might arise from this submission. Information about this grant can be found on the foundation website (https://www.gatesfoundation.org/about/committed-grants/2021/11/inv036678). The funders had no role in study design, data collection and analysis, decision to publish, or preparation of the manuscript.

**Competing interests:** The authors have declared that no competing interests exist.

>75% agreement among experts were considered as final, while the remaining were reviewed again by experts from UCD and MNF until consensus was reached on harmonized definitions. The current lexicon is available online at the DInA-website. and contains 217 terms and will be maintained as a "living document". The lexicon will facilitate the ability of key stakeholders of micronutrient programs to evaluate and compare program performance in order to make informed decisions on how to ensure future progress in reducing micronutrient deficiencies.

## Introduction

Micronutrient deficiencies (MNDs) are highly prevalent worldwide, and according to a recent pooled analysis, one in two preschool-aged children (equivalent to 372 million children) and two of three women of reproductive age (equivalent to 1.2 billion individuals) are deficient in at least one micronutrient [1]. The magnitude of the problem may be increasing due to disruptions in food supply chains and decreased access to nutritious foods imposed by climate change, the aftermath of the COVID-19 pandemic [2], and conflicts, including the ongoing war in Ukraine, a major global source of agricultural commodities [3]. MNDs place preschool children at an increased risk of growth stunting, blindness, impaired cognitive functions, elevated infectious disease, morbidity and mortality. Pregnant women with MNDs are at risk of poor pregnancy outcomes and increased maternal mortality [4–6]. People living in low- and middle-income countries are the most vulnerable to MNDs because safe and nutritious foods are neither accessible nor affordable for many families [7], while those in high-income countries are also affected [1, 8].

Well-known strategies to improve the access to and intake of micronutrients include promoting and protecting breastfeeding, improving dietary diversity, biofortification, large scale food fortification (LSFF), and supplementation [4, 8–10]. Despite the importance of micronutrient malnutrition and related intervention strategies, there is a dearth of accurate, representative, and timely data on population micronutrient status [11]. One impediment to more effective, multisectoral communication and intervention is the complex and often confusing terminology used by the diverse stakeholders involved in micronutrient programs. For example, different terms, such as "reach", "coverage" and "effective coverage" are applied indiscriminately, and sometimes incorrectly, for the monitoring and evaluation of micronutrient intervention programs [12]. These terms should be clearly defined for better understanding and communication among different sectors. Among the different micronutrient interventions, LSFF requires involvement from several different sectors, including government/public sector, business/private sector, donor agencies, food industries, academic institutions, and social marketing agencies, as well as consumers. As a result, standardized usage of terms is challenging, but it is important to have a common vocabulary for the terms used in LSFF and other micronutrient programs to facilitate communication and mutual understanding.

The Micronutrient Forum hosts the Micronutrient Data Innovation Alliance (DInA) to address such information gaps and to improve data utilization among stakeholders throughout the micronutrient data value chain. One of the first activities of DInA was to develop a lexicon of terms relevant to micronutrient status assessment and intervention programs. The purpose of this lexicon is to establish a central repository of key terms and their consensus definitions related to micronutrient programs with the overarching goals of facilitating communication among stakeholders, fostering effective policy- and decision-making, supporting design,

implementation, and monitoring and evaluation of programs, and facilitating comparison of performance between micronutrient programs. The present manuscript describes the methods used to arrive at the present consensus definitions of key terms in the lexicon.

## Methods

To identify a broad list of potentially relevant terms, an initial list of terms related to micronutrient and LSFF programs was prepared by searching multiple relevant sources, including reference documents from United Nations (UN) organizations, such as the World Health Organization (WHO), United Nations International Children's Emergency Fund (UNICEF), the Food and Agriculture Organization (FAO), and the World Food Programme (WFP) websites; multi-lateral governmental and non-governmental technical support agencies, such as the United States Agency for International Development (USAID), the Global Alliance for Improved Nutrition (GAIN), and the Program for Appropriate Technology in Health (PATH); websites relevant to micronutrients, such as the Global Fortification Data Exchange (GFDx) and OpeN-Global; and by searching the scientific and technical literature, including legal and regulatory literature. A full list of references currently used in the Lexicon can be found in **S1 File**. The terms search was made with the focus on different approaches of micronutrient interventions (fortification, supplementation, dietary diversification) and the phases of program cycles in which they are generally applied.

The definitions of terms together with the URL for original resources were compiled from the different sources mentioned above. To provide a clear framework for program management, and to ensure all aspects of the program are systematically addressed, the terms were categorized according to the programmatic phase, i.e., assessment, planning, implementation, monitoring and evaluation (**Table 1**). The terms compiled under the assessment phase/category include those related to assessment of food availability (sub-category food vehicle), dietary intake (sub-category diet), biochemical status (sub-category biochemical), anthropometry (sub-category anthropometry), industrial capacity (sub-category food fortification), and communication channels (sub-category food fortification). For the planning phase, the terms were assigned in relation to specific intervention strategies (sub-categories dietary diversification, fortification, supplementation, or all micronutrient interventions); intervention designs, such as mandatory or voluntary fortification (sub-category fortification); stages of the food chain in which nutrients are added to the food such as biofortification, industrial fortification, or home fortification (sub-category fortification). The terms allocated to the implementation phase are relevant to program delivery by public sector bodies, including operational capacity and efficiency, coverage, regulation, and compliance (sub-categories dietary diversification, fortification, supplementation, or all micronutrient interventions). The terms assigned to the monitoring and evaluation phase were listed under sub-category supply-side, such as service provider, suppliers and producers, or under sub-category demand-side such as target population or the consumers. As many terms are applicable to multiple phases of micronutrient programs, they were often classified under more than one category. The purpose of the categories is both to help structure the lexicon and to help users identify relevant terms. In addition, keywords were assigned for each term to facilitate users' searches, for example, the keyword "nutrient reference value" was applied for the terms "adequate intake", "dietary reference intake", "average requirement", "estimated average requirement", "tolerable upper intake level", etc.

The first draft of lexicon (n = 113 terms) was reviewed by experts from University of California, Davis (SYH, KHB), and Micronutrient Forum (RA, SJMO, MVL, MWB, LMH). For terms with definitions derived from multiple sources, all definitions were listed and checked

**Table 1. Intervention types and phases of program cycle considered during search for potential terms of interest for inclusion in lexicon.**

| Categories | Sub-categories |
|---|---|
| Intervention types | Mandatory fortification |
| | Voluntary fortification |
| | Biofortification |
| | Industrial fortification |
| | Home fortification |
| | Dietary diversification |
| | Supplementation |
| Assessment | Food availability |
| | Dietary intake |
| | Biochemical status |
| | Anthropometry |
| | Industrial capacity |
| | Communication channels |
| Planning | Dietary diversification |
| | Fortification |
| | Supplementation |
| | Micronutrient interventions |
| Implementation | Public sector program delivery |
| | Operational capacity |
| | Efficiency |
| | Coverage |
| | Regulation |
| | Compliance |
| Monitoring and evaluation | Supply-side (service provider, supplier, producer) |
| | Demand-side (target population, consumer) |

for consistency. A total of 115 terms were compiled in the second draft of lexicon, of which 54 terms had multiple definitions among different sources. Of these 54 terms, 12 terms had only a nominal difference in definitions among various sources and minor modification was made to harmonize them. The second draft (n = 115 terms) was then reviewed by Nutrition experts (n = 24) during a micronutrient and LSFF Stakeholder Alignment Convening and three Regional Consultations in June 2022. The experts who reviewed the second draft of Lexicon included those from the Bill and Melinda Gates Foundation, FAO, GAIN, Deutsche Gesellschaft für Internationale Zusammenarbeit (GIZ), Iodine Global Network (IGN), Johns Hopkins University, London School of Hygiene and Tropical Medicine, Micronutrient Forum, Nutrition International, UNICEF, University of California, Davis, and WFP.

After the second round of expert review, a modified Delphi approach was used to reach consensus on the remaining 42 terms with multiple definitions, and any terms for which modifications were proposed [13]. First, the draft lexicon was circulated to a large group of stakeholders with expertise in micronutrient status assessment and intervention implementation via three online survey forms (one form for 16 terms on food fortification; one form for 16 terms related to quality assurance, quality control and data ecosystem; and one form for 10 terms related to nutrients and biomarkers). The online survey along with access to an excel file with all 115 terms was also shared with experts via the Micronutrient Forum Newsletter, group mailing lists, and personal communication. A total of 39 respondents with diverse areas of expertise (10 experts in the area of large-scale food fortification, 10 experts in micronutrient

requirements, 13 experts in micronutrient surveys and research, two experts in policy development or legislation, food industry regulations, and food safety, one expert in program design or implementation, and three experts in public health nutrition) provided their input through the online survey. These experts represent diverse sectors including government (n = 9), academia (n = 18), non-profit or non-governmental organizations (n = 8), technical advisory services (n = 4); and were affiliated with various other institutions and organizations. The organizations of the experts who contributed to the review of Lexicon terms are listed in **S2 File**. For terms with multiple definitions, the respondents were asked to select the most appropriate and comprehensive definition or to propose a modified definition and to provide additional feedback, if any. Considering the lexicon to be a living document, additional micronutrient terms (n = 83 terms) were subsequently added to the Lexicon while the Delphi process is ongoing. Given the importance of safety and quality control in fortification and supplementation strategies and the need to enforce adherence to the standards and guidelines, 19 legal terms (those on policies, regulations, laws, acts, and legislation related to food fortification) were also added to the lexicon in consultation with a regulatory expert. These terms are important from the perspective of program delivery because the success of an LSFF program can be enhanced by an enabling legal and regulatory landscape in a country, therefore making it integral for stakeholders to understand these terms. The responses from the online survey were then summarized and terms with > 75% agreement among respondents were considered final for inclusion in the lexicon. All other terms were considered pending or provisional. The pending terms were then reviewed by the core contributors (SYH, KHB, MWB, RA, LMH, AG) to arrive at consensus definitions. The workflow of the lexicon development is illustrated in **Fig 1**.

## Results

The current version of the lexicon comprises 217 terms in total (**S1 Data**). Among them, there are 116 terms categorized under the category assessment, 91 under planning, 151 under implementation, 108 under monitoring and evaluation. As mentioned above, terms could be assigned to multiple categories, when appropriate. For example, the term "adequate intake" was categorized under "assessment" for dietary intake assessments as well as under "planning" and "implementation" to reflect its use in program planning and implementation to determine the desired intake level of specific nutrient of interest.

Of more than 140 experts who were sent the online survey forms regarding terms with multiple definitions through DInA newsletter, 39 participated in the online survey. These reviewers were experts in policy development, food industry regulations, and food safety as well as aspects of micronutrient assessment and program implementation. Among them, 16 experts addressed the terms related to food fortification, 19 addressed nutrient and biomarkers related terms, and 14 provided feedback on terms related to quality assurance, quality control and data ecosystem.

A web-based repository for the lexicon (https://dinalexicon.micronutrientforum.org/) was created to publish the terms and definitions along with a search function. The terms are sorted alphabetically and the total number of terms in the current version of lexicon can be seen on the first page. Users can search for specific terms by typing a keyword, or browse entries listed alphabetically and by program phases and sub-categories. Related terms can also be explored by clicking the initial item, which will take the users to a new tab. For example, the keyword "requirement" is linked to 19 related terms, including "apparent intake", "average requirement", "estimated average requirement", "dietary reference intake", and "recommended nutrient intake" among others.

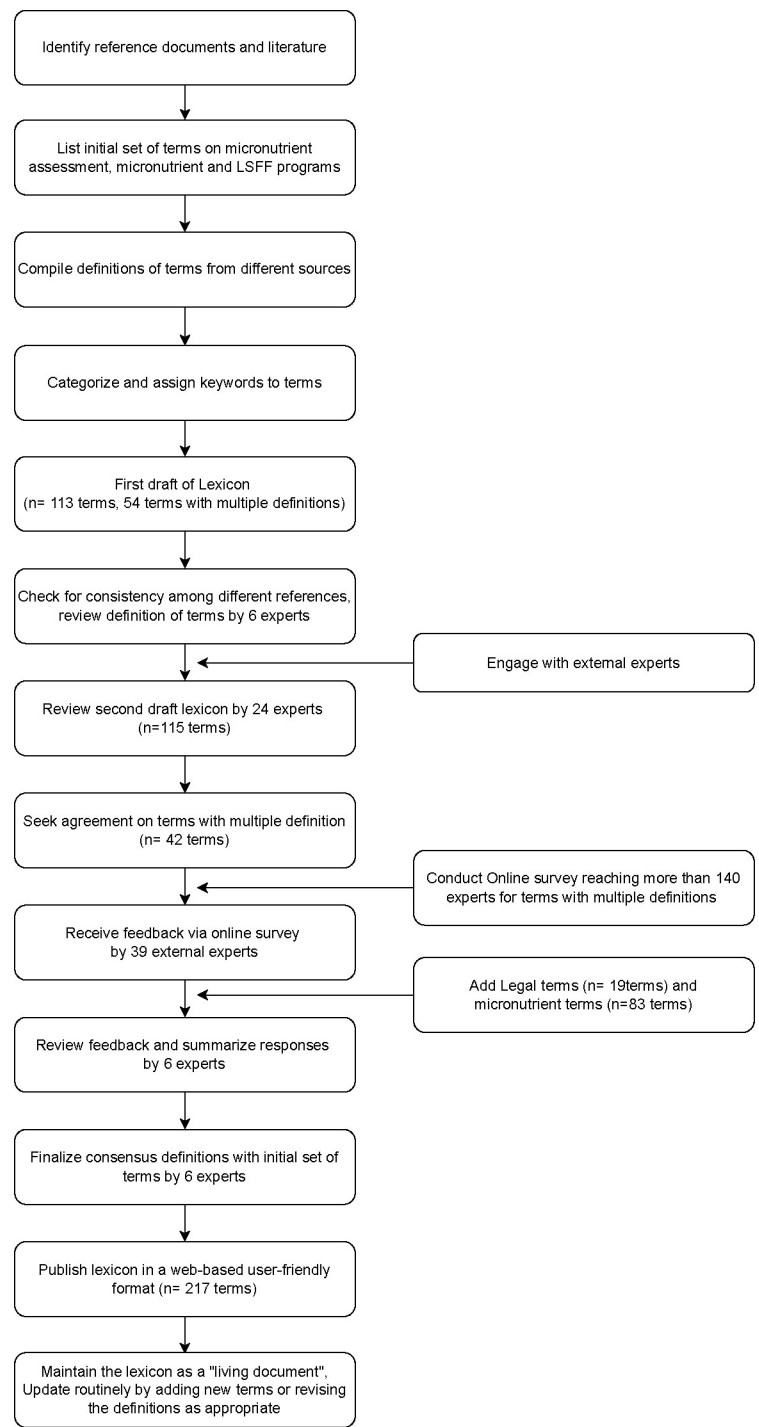

**Fig 1. Workflow for development of micronutrient and large-scale food fortification data lexicon.**

## Discussion and conclusion

The micronutrient and LSFF data lexicon is a central repository of terms used in micronutrient status assessment and intervention programs. Version one of the lexicon, containing 217 terms, is now published on the DInA website in a searchable and user-friendly format. The

lexicon is available to stakeholders in the micronutrient data landscape, including policy makers, donors, public sector implementing agencies, food industries, food producers and manufacturers, other relevant professionals in nutrition or adjacent sectors to facilitate communication regarding micronutrient program development including large scale food fortification programs. As one example of the utility of the lexicon, WHO is planning to use it as the basis for a glossary of terms to be included in the forthcoming revision of its publication, "Guidelines for Large Scale Food Fortification".

In establishing the lexicon, one of the challenges was to reach consensus on some of the definitions due to various perspectives among experts with different areas of expertise. However, this also highlights the need for a common lexicon and the importance of bringing together experts across various sectors or experts with different perspectives to reach consensus. Another limitation was the low response rate (~28%) to the online survey: only 39 experts out of more than 140 who were provided the online survey forms through DInA newsletter and personal communications participated in the online survey. This response rate of 28% is below the average response rates of online surveys summarized in a recent meta-analysis [14], which may have implications on the types of terms and respective definitions included in the lexicon due to a limited expertise and viewpoints. Particularly, we were unable to obtain input from food manufacturers during the consensus building process of the current lexicon. Therefore, despite multiple outreach strategies, it was not possible to reach experts who represent all stakeholder groups and global perspectives for each term, so it is not possible to claim that that the definitions represent a truly global consensus.

To ensure that the lexicon maintains its relevance and use throughout the micronutrient data value chain and over time, the lexicon will be maintained as a living online platform, which will be updated with new terms and/or with improved definitions as they are developed through new references and/or following suggestions from the stakeholder community. The potential need for adding more terms and respective definitions to the online lexicon will be reviewed every two years or as needed based on the updated guideline. The DInA team will also continue to be open to input and feedback to further improve the lexicon. The above-described modified Delphi approach of consensus building will be implemented. Future online surveys will be sent out to the growing DInA network. In an effort to increase the response rate, key experts will be pre-contacted prior to the email blast and followed up by personal email to encourage responsiveness as recently suggested by Wu et al. [14].

By offering a comprehensive and standardized set of terms, the lexicon can play a crucial role in improving communication across multi-disciplinary teams working in the field of micronutrient-related programs worldwide. While encouraging the universal adoption of standardized definitions may pose some challenges, the lexicon serves as a collaborative foundation, offering a shared consensus starting point and readily accessible definitions for key terms in micronutrient programs. The use of consensus definitions will be essential for key stakeholders of micronutrient programs to evaluate and compare program performance in order to make informed decisions on how to ensure future progress in reducing micronutrient deficiencies.

## Supporting information

**S1 File. List of references for terms search in Lexicon.**
(DOCX)

**S2 File. List of organizations of the experts who contributed to the review of Lexicon terms.**
(DOCX)

**S1 Data. All terms of Lexicon with their definitions and sources.**
(XLSX)

## Acknowledgments

The authors acknowledge all the experts who have participated in the online survey, who have provided their feedback and inputs on the process of lexicon development and definitions of terms in the lexicon. The authors specifically thank: Álvaro Pérez, Edward Joy, Eleanor Brible, Helena Pachon, Khalilur Rahman, Kristine Montecillo, Lindsay H Allen, Ludmila Ivanova, Mari Skar Manger, Omolara Ibiwumi Okunlola, Philip Randall, Quentin Johnson, Ryan Wessells, Savitesh Kushwaha, and Suzanne Fuhrman for their participation in the modified-Delphi feedback rounds.

## Author Contributions

**Conceptualization:** Kenneth H. Brown, Reed Atkin, Saskia J. M. Osendarp, Sonja Y. Hess.

**Data curation:** Lwin Mar Hlaing, Megan W. Bourassa.

**Investigation:** Kenneth H. Brown, Sonja Y. Hess.

**Methodology:** Lwin Mar Hlaing, Kenneth H. Brown, Sonja Y. Hess.

**Project administration:** Lwin Mar Hlaing, Megan W. Bourassa.

**Resources:** Lwin Mar Hlaing, Reed Atkin.

**Supervision:** Megan W. Bourassa.

**Writing – original draft:** Lwin Mar Hlaing.

**Writing – review & editing:** Lwin Mar Hlaing, Megan W. Bourassa, Kenneth H. Brown, Reed Atkin, Saskia J. M. Osendarp, Aishani Gupta, Sonja Y. Hess.

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
