## [Decision Letter · Decision Letter 0]

29 Apr 2024

PONE-D-24-04315Development of a standardized consensus lexicon for terms related to micronutrient programsPLOS ONE

Dear Dr. Hlaing,

Thank you for submitting your manuscript to PLOS ONE. After careful consideration, we feel that it has merit but does not fully meet PLOS ONE’s publication criteria as it currently stands. Therefore, we invite you to submit a revised version of the manuscript that addresses the points raised during the review process. Reviewer 1 has some practical suggestions for improving the exposition. Reviewer 1 also comments that omitting the private sector from the stakeholder group is not ideal.

We look forward to receiving your revised manuscript.

Kind regards,

Susan Horton

Academic Editor

PLOS ONE

Journal Requirements:

"This work was supported by the Bill & Melinda Gates Foundation (Grant number of INV-036678). Under the grant conditions of the Foundation, a Creative Commons Attribution 4.0 Generic License has already been assigned to the Author Accepted Manuscript version that might arise from this submission. Information about this grant can be found on the foundation website (https://www.gatesfoundation.org/about/committed-grants/2021/11/inv036678)."

**Additional Editor Comments:**

The reviewers agree that this is a useful and important paper. Reviewer 1 has some suggestions for minor improvements.

Reviewers' comments:

Reviewer's Responses to Questions

**Comments to the Author**

1. Is the manuscript technically sound, and do the data support the conclusions?

Reviewer #1: Yes

Reviewer #2: Yes

2. Has the statistical analysis been performed appropriately and rigorously? 

Reviewer #1: N/A

Reviewer #2: N/A

3. Have the authors made all data underlying the findings in their manuscript fully available?

Reviewer #1: Yes

Reviewer #2: Yes

4. Is the manuscript presented in an intelligible fashion and written in standard English?

Reviewer #1: Yes

Reviewer #2: Yes

5. Review Comments to the Author

Reviewer #1: This manuscript provides a valuable resource in standardizing terminology across micronutrient programs. Below are my key points and recommendations for improvement.

Methodology

The methodology section is well-explained but could emphasize the criteria for term selection and the process of consensus-building among experts. This would strengthen the rigor and transparency of the lexicon development process.

A key weakness in the methodology is the apparent absence of food manufacturers/processors (of fortified food) and food regulators and/or food safety experts in the list of experts who reviewed the Lexicon of terms. These stakeholders often rely on precise definitions of terms to develop regulatory standards and to comply with these standards.

Results and Discussion

The results are clearly presented, showing the extensive work done to compile and refine the lexicon. The discussion highlights the lexicon's potential impact but should address the low response rate in the expert surveys more critically, discussing its implications on the representativeness and global consensus of the definitions.

Lines 174-185: The presentation of data is clear, but it could be enhanced by discussing the implications of these findings. For instance, what does the categorization of terms indicate about the focus areas in micronutrient programs? Also, consider presenting the data in a table for easier comprehension and reference.

Lines 195-225: The discussion effectively highlights the lexicon's importance and potential impact. However, the challenges and limitations mentioned could be better framed as opportunities for future engagement and development. For example, the low response rate could be addressed by discussing strategies to increase participation and representation in future updates.

Line 216-219: The commitment to maintaining the lexicon as a living document is commendable. Expanding on how this will be achieved, including the process for incorporating new terms and updating definitions, would be beneficial.

General Recommendations

1. Provide a clearer rationale for categorizing terms and how these categories aid the lexicon’s usability for different stakeholders.

• Emphasize the impact of the lexicon in the discussion, particularly how it can enhance communication and decision-making in micronutrient programs. Discuss the real-world application of the lexicon, such as how it has been used in policy-making, program design, or academic research, to demonstrate its practical value.

The manuscript represents a significant effort in standardizing terminology in the field of nutrition and has the potential to be a vital resource for stakeholders in micronutrient programs. With the recommended improvements, I believe it will make a substantial contribution to the literature. However, as mentioned above, the absence of manufacturers of fortified foods and food regulators and/or food safety experts concerns me.

Reviewer #2: This is a very useful effort to standardize the lexicon and terminology used by different stakeholders working to address micronutrient deficiencies. The methodology used is sound and I recommend the publication of the paper without any modifications.

6. PLOS authors have the option to publish the peer review history of their article (what does this mean?). If published, this will include your full peer review and any attached files.

Reviewer #1: **Yes: **Rolf DW Klemm

Reviewer #2: **Yes: **Marthi Venkatesh Mannar

---

## [Author Response · Author response to Decision Letter 0]

15 Jul 2024

We thank the editors and reviewers for their time to review the manuscript (ID: PONE-D-24-04315) and provide valuable feedback for improvement. 

Response to Editor's comments

1. We have formatted the manuscript to ensure meeting the PLOS ONE's style requirements.

2. Financial disclosure statement has also been revised as suggested. "This work was supported by the Bill & Melinda Gates Foundation (Grant number of INV-036678). Under the grant conditions of the Foundation, a Creative Commons Attribution 4.0 Generic License has already been assigned to the Author Accepted Manuscript version that might arise from this submission. Information about this grant can be found on the foundation website (https://www.gatesfoundation.org/about/committed-grants/2021/11/inv036678). The funders had no role in study design, data collection and analysis, decision to publish, or preparation of the manuscript." The statement is also added in the Cover letter.

3. Data availability statement has been edited as "All relevant data are within the manuscript and its Supporting Information files" as all our data has been provided in the Supporting Information files. In addition, all data (Micronutrient-related terms with definitions in the current lexicon) can be accessible in the Micronutrient Forum website through https://dinalexicon.micronutrientforum.org/

4. This manuscript describes the methodology of achieving the consensus definitions of key terminologies related to different micronutrient programs. Therefore, it is not relevant to Ethical statement. Accordingly, we did not mention about Ethics statement in the "Methods" section of the Manuscript. We have also mentioned about it in the "Human Subjects Research Checklist" document.

5. We have added the captions for the Supporting Information files at the end of the manuscript.

Responses to Reviewers' comments

Please see below our point-by-point response to the comments from reviewers. The revisions are made with track changes in the manuscript file, and we also submitted an unmarked version of the revised manuscript. The line numbers correspond to the marked version. We have also attached the "Responses to reviewers" document in the submission. We hope that you will find the revisions to the manuscript thorough and satisfactory. 

Reviewer #1: 

This manuscript provides a valuable resource in standardizing terminology across micronutrient programs. Below are my key points and recommendations for improvement.

Methodology

The methodology section is well-explained but could emphasize the criteria for term selection and the process of consensus-building among experts. This would strengthen the rigor and transparency of the lexicon development process.

We have added the description that the selection and categorization of terms was based on different approaches of micronutrient interventions (fortification, supplementation, dietary diversification) and the phases of program cycles, i.e., assessment, planning, implementation, monitoring and evaluation (lines 30-32, 111-113, and 115-116).

A key weakness in the methodology is the apparent absence of food manufacturers/processors (of fortified food) and food regulators and/or food safety experts in the list of experts who reviewed the Lexicon of terms. These stakeholders often rely on precise definitions of terms to develop regulatory standards and to comply with these standards.

We have circulated the online survey forms to solicit the opinions to a larger group of experts (more than 140 in number) including experts in food legislation and food safety. There were experts in policy development, food industry regulations, and food safety among the 39 experts who have responded to the online survey. Legal terms were also added in the lexicon in consultation with a legal expert, who contributed as coauthor to this manuscript. We have added further details to the methods section (lines 166-167, 180) and results section (lines 199-201). Despite the above-described efforts, we were not able to obtain feedback from food manufacturers in the consensus building process of the current lexicon. We have added this in the discussion (lines 234-236 and lines 247-249) as potential weakness and for consideration when adding new terms to the lexicon.

Results and Discussion

The results are clearly presented, showing the extensive work done to compile and refine the lexicon. The discussion highlights the lexicon's potential impact but should address the low response rate in the expert surveys more critically, discussing its implications on the representativeness and global consensus of the definitions.

Thank you for this comment. The discussion on the low response rate of expert survey has been edited as suggested (lines 231-236).

Lines 174-185: The presentation of data is clear, but it could be enhanced by discussing the implications of these findings. For instance, what does the categorization of terms indicate about the focus areas in micronutrient programs? Also, consider presenting the data in a table for easier comprehension and reference.

As suggested, we have added Table 1 (line 140) to provide an overview of the categories and sub-categories that guided the search for terms. We also clarified that rationale for considering these terms as follows on lines 30-32, 111-113, and 115-116: “To provide a clear framework for program management, and to ensure all aspects of the program are systematically addressed, the terms were categorized according to the programmatic phase,…”

Lines 195-225: The discussion effectively highlights the lexicon's importance and potential impact. However, the challenges and limitations mentioned could be better framed as opportunities for future engagement and development. For example, the low response rate could be addressed by discussing strategies to increase participation and representation in future updates.

We have added this to the discussion lines 231-236 and 247-249. 

Line 216-219: The commitment to maintaining the lexicon as a living document is commendable. Expanding on how this will be achieved, including the process for incorporating new terms and updating definitions, would be beneficial.

The discussions on the process of incorporating new terms and updating definitions, along with a plan to increase participation and representation in future updates are added on lines 242-249. 

General Recommendations

Provide a clearer rationale for categorizing terms and how these categories aid the lexicon’s usability for different stakeholders.

As mentioned above, we have added the description on lines 30-32, 111-113, and 115-116 and adding Table 1.

Emphasize the impact of the lexicon in the discussion, particularly how it can enhance communication and decision-making in micronutrient programs. Discuss the real-world application of the lexicon, such as how it has been used in policy-making, program design, or academic research, to demonstrate its practical value.

The potential implications of the lexicon has been addressed in lines 250-252.

The manuscript represents a significant effort in standardizing terminology in the field of nutrition and has the potential to be a vital resource for stakeholders in micronutrient programs. With the recommended improvements, I believe it will make a substantial contribution to the literature. However, as mentioned above, the absence of manufacturers of fortified foods and food regulators and/or food safety experts concerns me.

Thank you very much again for your valuable comments and suggestions to improve our manuscript. The critical discussion and future plan to include food manufacturers when adding new terms was described on lines 234-236 and lines 246-252. 

Reviewer #2: 

This is a very useful effort to standardize the lexicon and terminology used by different stakeholders working to address micronutrient deficiencies. The methodology used is sound and I recommend the publication of the paper without any modifications.

Thank you very much for your review and recommendation for publication.

---

## [Editor Report · Decision Letter 1]

19 Jul 2024

Development of a standardized consensus lexicon for terms related to micronutrient programs

PONE-D-24-04315R1

Dear Dr. Hlaing,

We’re pleased to inform you that your manuscript has been judged scientifically suitable for publication and will be formally accepted for publication once it meets all outstanding technical requirements.

Kind regards,

Susan Horton

Academic Editor

PLOS ONE
---

## [Editor Report · Acceptance letter]

23 Jul 2024

PONE-D-24-04315R1 

PLOS ONE

Dear Dr. Hlaing, 

I'm pleased to inform you that your manuscript has been deemed suitable for publication in PLOS ONE. Congratulations! Your manuscript is now being handed over to our production team.

Kind regards, 

on behalf of

Dr. Susan Horton 

Academic Editor

PLOS ONE